# Targeting human langerin promotes HIV-1 specific humoral immune responses

Jérôme Kervevan[1], Aurélie Bouteau[2,3,4☯], Juliane S. Lanza[5,6☯], Adele Hammoudi[1], Sandra Zurawski[2], Mathieu Surenaud[1], Lydie Dieudonné[1], Marion Bonnet[7], Cécile Lefebvre[1], Hakim Hocini[1], Romain Marlin[8], Aurélie Guguin[9], Barbara Hersant[10], Oana Hermeziu[10], Elisabeth Menu[8,11], Christine Lacabaratz[1], Jean-Daniel Lelièvre[1], Gerard Zurawski[2], Véronique Godot[1], Sandrine Henri[6], Botond Z. Igyártó[2,4], Yves Levy[1,12☯*], Sylvain Cardinaud[1☯*]

1 Vaccine Research Institute, Créteil, France, Inserm U955, Équipe 16, Créteil, France, 2 Baylor Institute for Immunology Research (BIIR), Vaccine Research Institute, Dallas, Texas, United States of America, 3 Institute of Biomedical Studies, Baylor University, Waco, Texas, United States of America, 4 Thomas Jefferson University, Department of Microbiology and Immunology, Philadelphia, Pennsylvania, United States of America, 5 Centre d'Immunophénomique, Aix Marseille Université, INSERM, CNRS, Marseille, France, 6 Centre d'Immunologie de Marseille-Luminy, Aix Marseille Université, INSERM, CNRS, Marseille, France, 7 College of Biomedical and Life Sciences, University of Cardiff, Cardiff, United Kingdom, 8 CEA, Université Paris-Sud, Inserm, U1184 "Immunology of Viral Infections and Autoimmune Diseases" (IMVA), IDMIT Department, IBFJ, Fontenay-aux-Roses, France, 9 Inserm U955 –Plateforme de Cytométrie, Institut Mondor de Recherche Biomédicale, UPEC, Créteil, France, 10 Department of plastic and maxillo-facial surgery, Henri Mondor Hospital, Créteil, France, 11 MISTIC Group, Department of Virology, Institut Pasteur, Paris, France, 12 AP-HP, Hôpital Henri-Mondor Albert-Chenevier, Service d'Immunologie Clinique et Maladies Infectieuses, Créteil, France

☯ These authors contributed equally to this work.
* yves.levy@inserm.fr (YL); sylvain.cardinaud@inserm.fr (SC)

**Data Availability Statement:** All relevant data are within the manuscript and its Supporting Information files.

## Abstract

The main avenue for the development of an HIV-1 vaccine remains the induction of protective antibodies. A rationale approach is to target antigen to specific receptors on dendritic cells (DC) via fused monoclonal antibodies (mAb). In mouse and non-human primate models, targeting of skin Langerhans cells (LC) with anti-Langerin mAbs fused with HIV-1 Gag antigen drives antigen-specific humoral responses. The development of these immunization strategies in humans requires a better understanding of early immune events driven by human LC. We therefore produced anti-Langerin mAbs fused with the HIV-1 gp140z Envelope (αLC. Env). First, we show that primary skin human LC and *in vitro* differentiated LC induce differentiation and expansion of naïve CD4+ T cells into T follicular helper (Tfh) cells. Second, when human LC are pre-treated with αLC.Env, differentiated Tfh cells significantly promote the production of specific IgG by B cells. Strikingly, HIV-Env-specific Ig are secreted by HIV-specific memory B cells. Consistently, we found that receptors and cytokines involved in Tfh differentiation and B cell functions are upregulated by LC during their maturation and after targeting Langerin. Finally, we show that subcutaneous immunization of mice by αLC.Env induces germinal center (GC) reaction in draining lymph nodes with higher numbers of Tfh cells, Env-specific B cells, as well as specific IgG serum levels compared to mice immunized with the non-targeting Env antigen. Altogether, we provide evidence that human LC properly targeted may be licensed to efficiently induce Tfh cell and B cell responses in GC.

**Funding:** This work was financially supported by the Programme Investissements d'Avenir (PIA) managed by the Agence Nationale de Recherche (ANR) under reference ANR-10-LABX-77-01 (Labex VRI) (YL, SC, JK, JSL, LD). The National Institute of Health (NIH) (NIH R01AI146420) and the Baylor Foundation (institutional support) also supported this work (BZI, AB). JK., JSL and LD received salary from the ANR (ANR-10-LABX-77). The funders had no role in study design, data collection and analysis, decision to publish, or preparation of the manuscript.

**Competing interests:** I have read the journal's policy and the authors of this manuscript have the following competing interests: GZ, SZ, and YL are named inventors on patent applications held by Inserm and BIIR concerning Langerin targeting. All other authors have declared that no competing interest exist.

**Abbreviations:** HIV, human immunodeficiency virus; NHP, non-human primate; DC, dendritic cell; LC, Langerhans cells; CD34-LC, *in vitro* Langerhans cells generated from CD34+ hematopoietic stem cells; MoDC, monocyte-derived DC; PBMC, peripheral blood mononuclear cells; Tfh, follicular helper T cell; CTL, cytotoxic T lymphocyte; GC, germinal center; dLN, draining lymph node; Ab, antibody; mAb, monoclonal antibody; bNAbs, broadly neutralizing antibody; Ig, immunoglobulin; sc, subcutaneous; IP, intraperitoneal; Untx, untreated.

## Author summary

In recent years, the place of innovative vaccines based on the induction/regulation and modulation of the immune response with the aim to elicit an integrated T- and B cell immune responses against complex antigens has emerged besides "classical" vaccine vectors. Targeting antigens to dendritic cells is a vaccine technology concept supported by more than a decade of animal models and human pre-clinical experimentation. Recent investigations in animals underscored that Langerhans cells (LC) are an important target to consider for the induction of antibody responses by DC targeting vaccine approaches. Nonetheless, the development of these immunization strategies in humans remains elusive. We therefore developed and produced an HIV vaccine candidate targeting specifically LC through the Langerin receptor. We tested the ability of our vaccine candidate of targeting LC from skin explant and of inducing *in vitro* the differentiation of T follicular helper (Tfh) cells. Using complementary *in vitro* models, we demonstrated that Tfh cells induced by human LC are functional and the targeting of LC by our vaccine candidate promotes the secretion of anti-HIV IgG by memory B cells from HIV-infected individuals. In this study human LC exhibit key cellular functions able to drive potent anti-HIV-1 humoral responses providing mechanistic evidence of the Tfh- and B cell stimulating functions of primary skin targeted LC. Finally, we demonstrated in *Xcr*1^DTA^ mice the significant advantage of LC targeting for inducing Tfh and germinal center (GC)-B cells and anti-HIV-1 antibodies. Therefore, the targeting of the human Langerin receptor appears to be a promising strategy for developing efficient HIV-1 vaccine.

## Introduction

Thirty-five years after the beginning of the HIV-1 pandemic, the need for a globally effective HIV-1 vaccine is more compelling than ever. In the HIV vaccine clinical trial "RV-144", a modest protection against acquisition has been observed [1]. Results evoked that the protection could be achieved in the absence of neutralizing antibodies and was associated with the generation of anti-Env antibodies binding to the V1V2 regions, the IgG isotype and the antibody function. However, protection rapidly wane over time thus underscoring that the vaccine did not induce durable protective immune responses. Unfortunately, these promising results were not confirmed in the recent HIV vaccine prophylactic trial "HVTN 702", adapted from vaccine strategy tested in the RV-144 study. Thus, better vaccine designs or new concepts must be explored for developing effective vaccination regimens.

In recent years, the place of innovative vaccines based on the most recent knowledge of the induction/regulation and modulation of the immune response with the aim to elicit an integrated T and B cell immune responses against complex antigens has emerged beyond "classical" vaccine vectors (recombinant viruses, naked DNA or long peptides). Targeting antigens to endogenous DC appears as a promising strategy to reprogram the immune system [2]. The current approach is to use a specific monoclonal antibody directed against a particular endocytic receptor to carry the antigen to the DC, resulting in processing and presentation of antigens but also in the activation of DC depending on the targeting. Targeting antigens to DC is a vaccine technology concept supported by more than a decade of animal model and human pre-clinical experimentation. *In vitro* targeting of HIV-1 antigens to DC receptors (e.g. DEC-205, CD40, DCIR, LOX-1) on human PBMCs, or in animal models, induces, even with minute amounts of antigens, strong and sustained T- and B cell responses, associated with control of HIV-1 infection [3–9].

Despite extensive research over the past years, it remains difficult to define strategies that initiate Tfh cell differentiation, particularly in vaccination [10, 11]. LC are arrayed at barrier sites of foreign antigen insult and were initially reported as preferentially selecting and expanding antigen-specific cytotoxic T lymphocytes (CTL) [12]. Yet, our group and others have shown that targeting HIV or Flu antigens to LC elicited T and B cell responses [13–18]. We recently demonstrated that delivering a foreign antigen to steady state LC versus cDC1 through the same mouse Langerin receptor led to, respectively, strong versus minimal humoral immune responses [17]. Moreover, LC, unlike cDC1, support the formation of GC-Tfh in a dose-dependent manner. Altogether, these investigations underscore that LC are an important target to consider for the induction of antibody responses by DC targeting vaccine approaches.

The further development of these immunization strategies in humans requires a better understanding of early immune events driven by targeted human LC. We previously set up a pioneering *in vitro* model of human LC and demonstrated that in a fully autologous system, human LC are prompted to differentiate naïve CD4$^+$ T cell into Tfh-like cells [17]. Here we designed anti-Langerin mAbs fused with the HIV-1 Envelope protein and investigated *in vitro* and *in vivo* whether the targeting of the human Langerin receptor impacts the Tfh/B cell axis and induces the production of Env-specific immunoglobulins (Ig).

## Results

### Targeting HIV-1 Env to human LC increases IgG production by naïve B cells

We generated a fusion of HIV-1 Env to a previously described in-house anti-human Langerin human IgG4 mAb (αLC.Env) [18]. We asked whether targeting Env to human LC could promote Tfh/B cell functions. We first isolated primary epidermal LC and confirmed their phenotype by FACS staining (CD207$^+$ CD1a$^{hi}$ CD11c$^{low}$ CD1c$^+$) (**S1 Fig**). LC were treated with αLC. Env or IgG4.Env vaccine and cocultured them allogenic naïve CD4$^+$ T cells (**S2 Fig**). We first confirmed by staining of skin explant or dissociated epidermal cells that αLC.Env was binding Langerin$^+$ cells in a dose-dependent manner, whereas no binding of IgG4.Env was observed (**S3A and S3B Fig**). In coculture (**Fig 1A**), skin-isolated LCs induced differentiation of allogenic naïve CD4$^+$ T cells into Tfh cells (**Fig 1B**). To assess the functionality of Tfh cells, autologous naïve IgM$^+$ B cells were added to the culture. The treatment of epidermal LC with the αLC.Env vaccine construct significantly increased the production of total IgG ($P < 0.01$ and $P < 0.05$, as compared to the untreated or IgG4.Env non-targeted skin LC; respectively) (**Fig 1C**). We then investigated the differentiation of naïve B cells in a complete autologous system. Therefore, we differentiated *in vitro* LC from CD34$^+$ cord blood cells (CD34-LC), as previously described [17] (**Fig 2A**). Whilst CD34-LC were phenotyped by the typical markers of LC, the αLC.Env construct was binding to CD34-LC in a dose-dependent manner and was shown to be rapidly internalized (**S3C–S3E Fig**). We first confirmed that autologous naïve CD4$^+$ T cells isolated from the same cord blood donor differentiated spontaneously into Tfh-like cells at day 4 of coculture (average 67 ± 16%), without any addition of cytokines (**S4A and S4B Fig**). These cells were proliferating and ~10% of them were phenotyped as Tfr (CD25$^+$ FoxP3$^{hi}$). Whilst αLC.Env was captured by LC, this treatment did not significantly change the percentage of differentiated Tfh or Tfr cells (**S4C–S4E Fig**). Autologous naïve IgM$^+$ B cells were added to CD34-LC/CD4$^+$ T cell co-cultures at day 4, and B cells were phenotyped 6 days later. Strikingly, without any additional cytokine, 1.1% (± 0.1) of CD19$^+$ B cells were expressing CD27, a marker of memory B cells (**Fig 2B**). Most of them (94 ± 7%) were CD21$^+$, indicating a resting memory phenotype. Thus, Tfh cells differentiated by CD34-LC were functional and induced

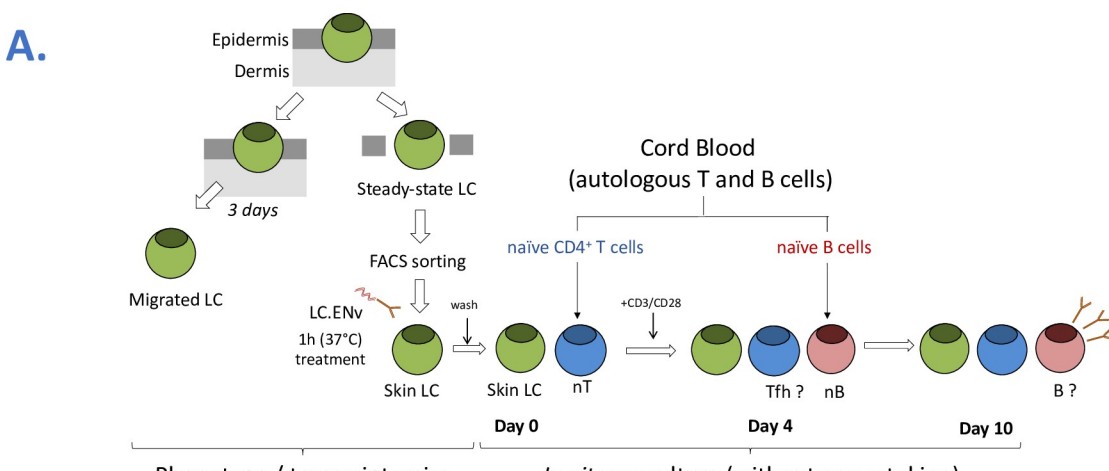

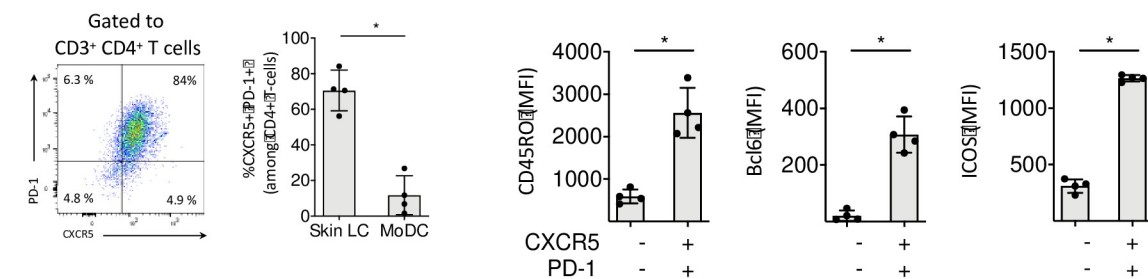

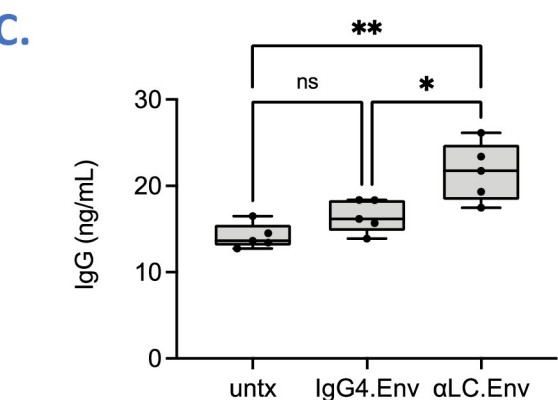

**Fig 1. Targeting human skin LC promotes IgG production by naïve B cells. (A)** Schematics of the procedure of the epidermal skin LC treated with αLC.Env (10nM) or control mAbs, cultured with naïve CD4+ T cells for 4 days and then with autologous naïve B cells for 6 days. **(B)** Human LC isolated from skin explants were cocultured with allogenic CD4⁺ naïve T cells as in A. Four days post culture, Tfh-differentiated cells were assessed by FACS, using CD45RO, CXCR5, PD-1, Bcl-6 and ICOS markers. DC were generated *in vitro* from CD14 + purified monocytes and cultured with skin LC as comparison. MFI: mean of fluorescence intensity. **(C)** IgG concentrations in LC/T/B co-culture supernatants were measured by ELISA at day 10 after LC treatment. The 10–1074 mAb with a known concentration served as reference. Statistics were obtained using the Holm-Sidak's multiple comparisons test (*, $P < 0.05$; **, $P < 0.01$; ***, $P < 0.001$; ns; non-significant).

## A.

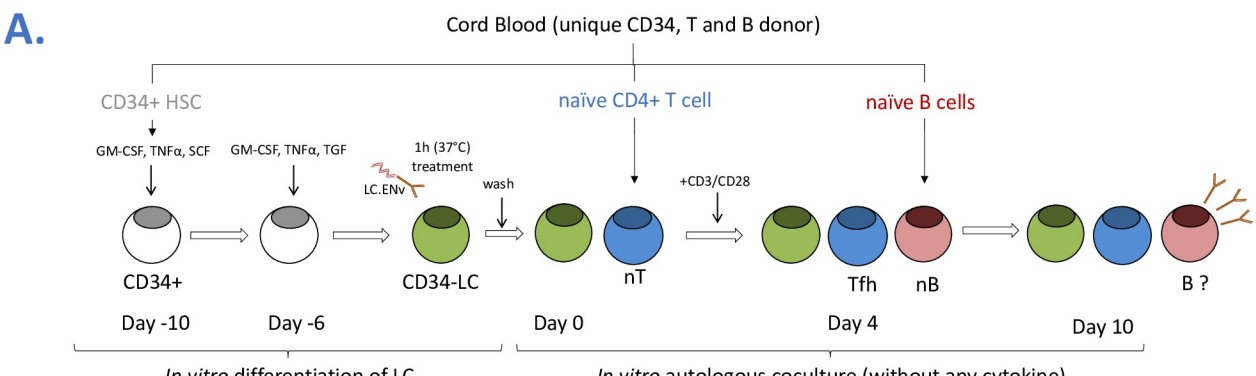

## B.

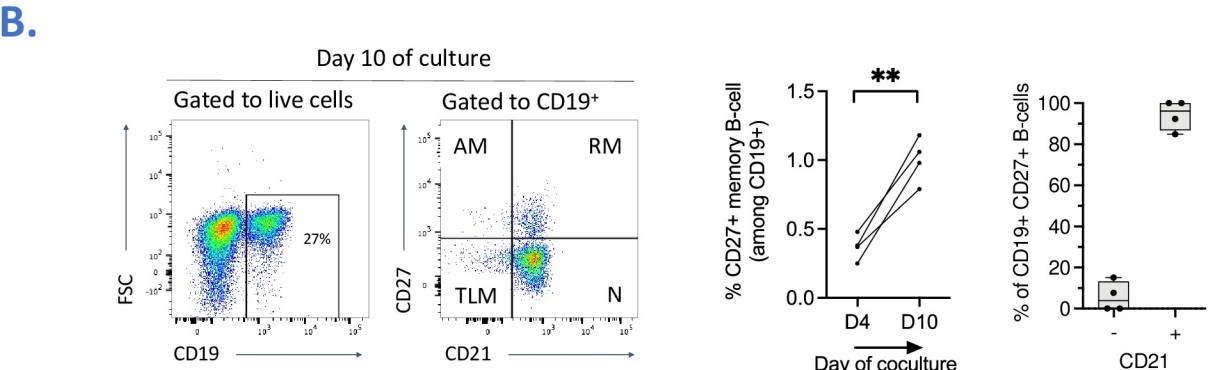

## C.

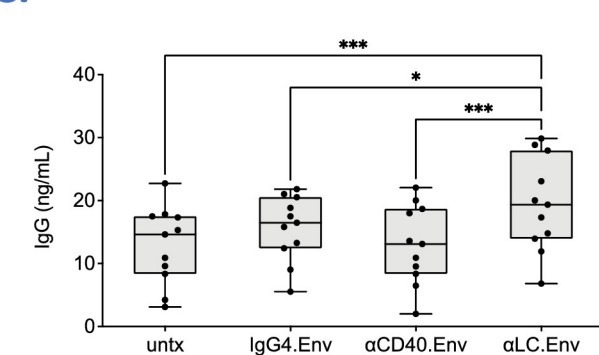

**Fig 2. Targeting of CD34-LC induces the activation of Tfh cells and Ig class-switch by naïve B cells, in a complete autologous system, without any cytokine. (A)** As in Fig 1A, schematics of the procedure of the CD34-LC treated with αLC.Env (10nM) or control mAbs, cultured with autologous Tfh cells and autologous naïve B cells for 6 days. **(B)** B cell phenotype was performed before and after coculture and was based on the expression of CD21 and CD27 among CD19$^+$ population. Representative dot plots of one out of four donors are depicted on the left (AM, activated memory, RM, resting memory; TLM; tissue-like memory; N, naïve). Two-tailed paired T-test, $^{**}$ $P < 0.01$ **(C)** As in Fig 1C, IgG concentrations in LC/T/B co-culture supernatants measured by ELISA at day 10 after LC treatment. Statistics were obtained using the Holm-Sidak's multiple comparisons test ($^*$, $P < 0.05$; $^{**}$, $P < 0.01$; $^{***}$, $P < 0.001$; ns; non-significant).

the differentiation of naïve B cells. Treating CD34-LC with αLC.Env significantly increased amounts of produced total IgG compared to non-targeted Env (IgG4.Env) ($P < 0.05$) or Env fused to an anti-CD40 receptor used as an irrelevant receptor (αCD40.Env) ($P<0.001$)

(**Fig 2C**). Taken together, these data provide evidence of a functional interaction between naïve B cells and co-cultured autologous Tfh/CD34-LC.

## Targeting Env to human CD34-LC induces the production of Env-specific Ig by B cells

The *in vitro* priming of HIV-Env B cells is limited by the frequency of circulating progenitors and the difficulty to reproduce the lymph node environment. Therefore, we decided to investigate whether the targeted human LC were able to stimulate the production of Env-specific antibodies by memory B cells. Autologous CD34-LC/Tfh cells were cultured with total blood B cells from allogenic HIV$^+$ donors, without any additional cytokine (**Fig 3A**). After 6 days, total secreted IgG was significantly increased when LC were treated with αLC.Env versus IgG4.Env or αCD40.Env ($P < 0.001$) (**Fig 3B, left panel**). No modulation in terms of IgG secreted by B cells was observed without adding naïve CD4$^+$ T cells at day 0, indicating a Tfh-dependent secretion of IgG (**Fig 3B, right panel**). A dose dependency between the amount of vaccine loaded on CD34-LC and the production of IgG was observed (**S5A Fig**). This IgG production was not modulated neither when CD34-LC were treated with *in-house* developed vaccine targeting the HIV-1 Gag protein to the human Langerin, nor when targeting the HIV-1 envelop to another lectin receptor (DCIR) (**S5B Fig**). Of note, we showed that the IgG4 detected at day 10 was not derived from our IgG4-based vaccine (**S5C Fig**). Strikingly, a significant production of Env-specific antibodies was observed when LC were treated with αLC.Env and cocultured with naïve CD4$^+$ T cells and allogenic B cells from HIV$^+$ donors, but not with neither IgG4. Env nor αCD40.Env control mAb (**Fig 3C**). Taken together, our results demonstrated that CD34-LC drive the differentiation of functional Tfh cells sustaining a production of both IgG and Env-specific Ig by B cells, and these were increased by targeting the antigen to LC, in a coculture system without addition of external cocktail of cytokines.

## Targeting skin LC promotes pro-Tfh / B cell functions

In order to decipher the functional profile of LC following αLC.Env targeting, skin LC were primed with either IgG4.Env or αLC.Env and then co-cultured with naïve CD4$^+$ T cells and allogenic total B cells from HIV$^+$ donors. Culture supernatants were assessed for cytokine production. In supernatants of skin LC cultured without CD4$^+$ T cells (**Fig 4A**), we detected minute amounts of cytokines, notably IL-22 and APRIL with medians at 15 and 35 pg/mL, respectively. Vaccine treatments did not induce any specific cytokine production. The BAFF and APRIL TNF family members are known to act on B cell functions [19]. In the presence of naïve CD4$^+$ T cells, APRIL was slightly more abundant, but BAFF remained undetectable. Th1-like cytokines (IFN-γ, TNFα, IL-2), IL-6 and GM-CSF were detectable, respectively. Cytokine production was in fact markedly increased in the presence of total B cells, with high amounts of GM-CSF, IL-13, IL-6, CXCL13 and APRIL (medians at 2628, 718, 658, 215 and 253 pg/mL, respectively). Noticeable, IL-21, a Tfh-derived cytokine which is also linked to B cell functions, was found in culture supernatants (**Fig 4B**). IL-18 and CXCL13 increased when skin LC were treated by αLC.Env (median > 2x untreated samples), though no significant difference could be observed with IgG4.Env (**Fig 4C**). We finally compared cytokines produced when culturing CD34-LC with autologous naïve CD4+ T cells and autologous naïve B cells or allogenic total B cells (**S6 Fig**). We observed similarities with skin LC in terms of cytokine profiles, particularly the high production of IL-6, IL-13, GM-CSF and IL-2. An IL-21 signature was also found in culture supernatants of Tfh cells (median at 53 pg/mL) and at increased levels in the presence of naïve or total B cells (medians at 133 and 142 pg/mL, respectively). As for skin LC, a significant production of APRIL, IL-18 and CXCL13 was markedly increased in the

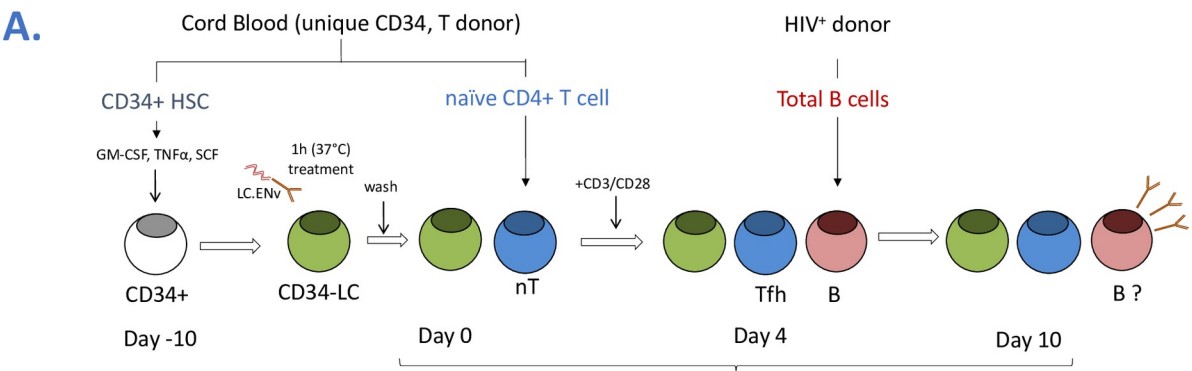

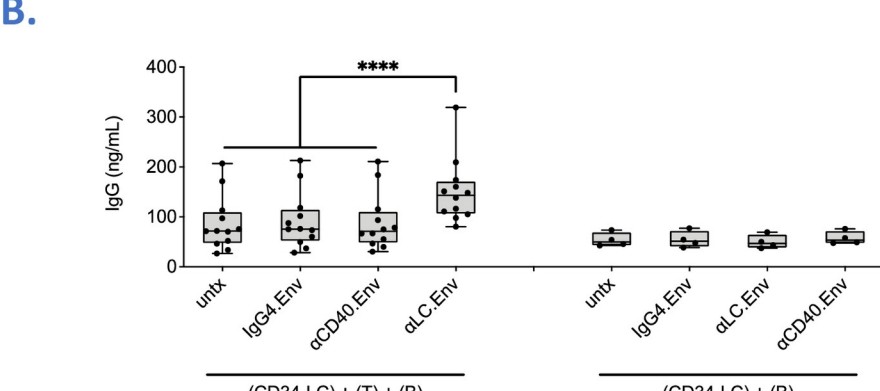

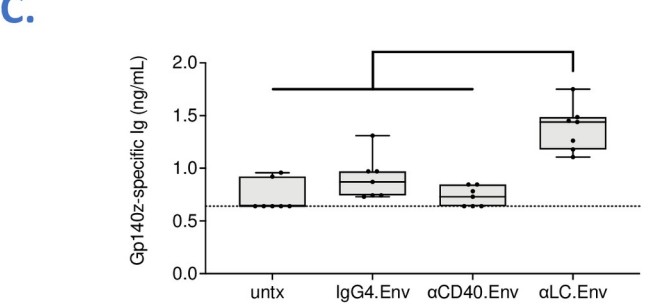

**Fig 3. Targeting CD34-LC induces the secretion of Env-specific Ig by memory B cells from HIV⁺ donors. (A)** Schematics of the procedure, as in Fig 2 but adding total B cells isolated from PBMCs of HIV⁺ donors at day 4. **(B)** As in Fig 2C, total IgG were measured in culture supernatants of LC treated or not with αLC.Env, and cultured (left) or not (right) with naïve CD4⁺ T cells. **(C)** ELISA-based concentration analysis of Env gp140z-specific Ig in culture supernatants at day 17. Statistics were obtained using the Holm-Sidak's multiple comparisons test (*, $P < 0.05$; **, $P < 0.01$; ***, $P < 0.001$; ns; non-significant).

presence of Tfh and B cells. Transcriptomic analysis of LCs isolated from different epithelia have shown that despite specific signatures induced by their microenvironment these cell subsets present gene expression homologies [20]. It is now well established that during their migration towards the draining lymph node, skin LC mature, with expression of cytokines and

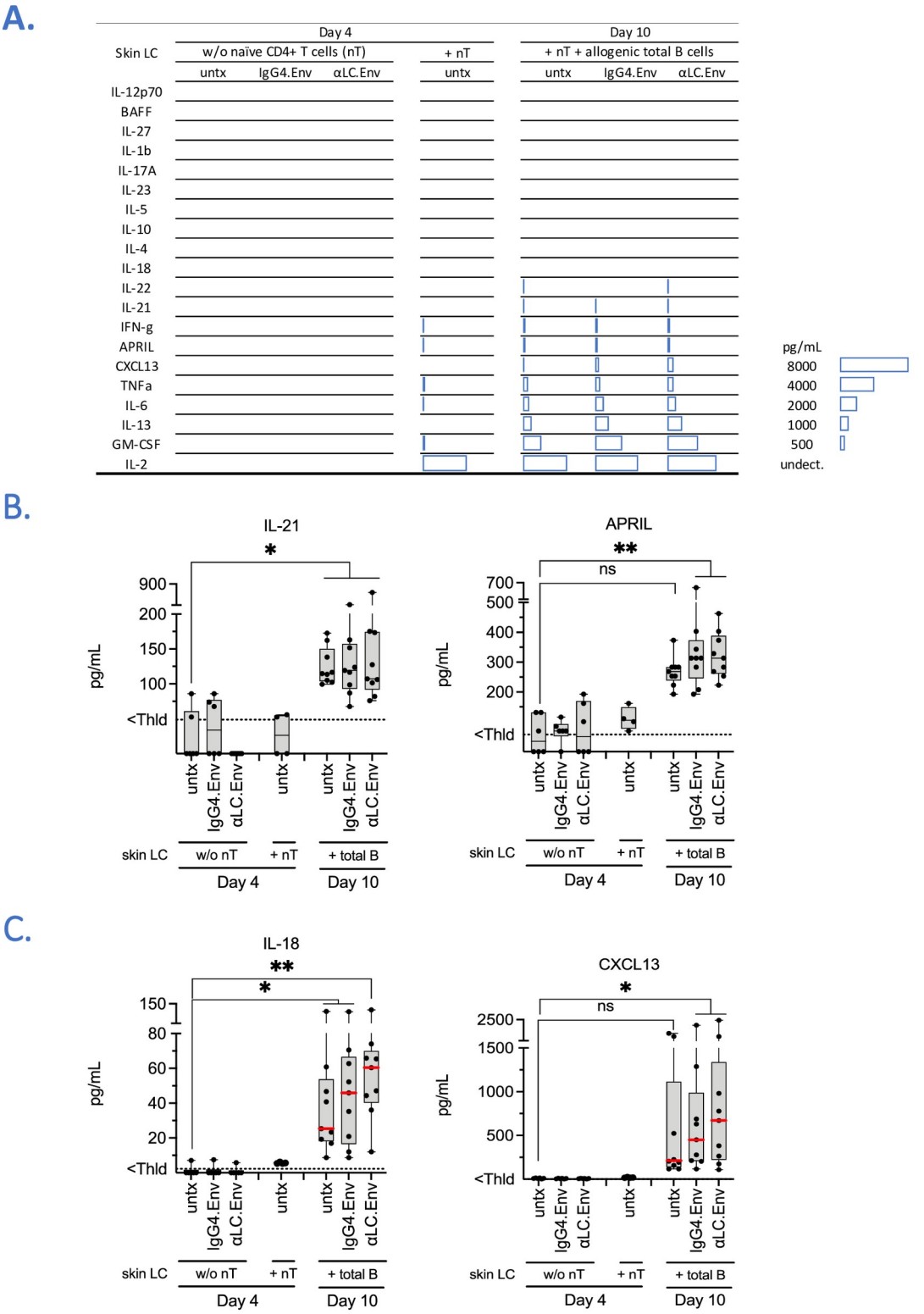

**Fig 4. Targeting skin LC promotes a specific cytokine environment. (A)** Multiplex beads analysis of 20 cytokines and comparison of culture supernatants at day 4 or 10 (as in Fig 1A) with skin LC treated by αLC.Env or IgG4.Env, cultured alone 4 days (left), with allogenic naïve CD4+ T cells (middle) or with total B cells from same donors (right). Color scale indicates median values of each condition (n = 4

to 9) and is expressed in pg/mL. Values of IL-21 and APRIL are detailed in (**B**), and IL-18 and CXCL13 in (**C**), respectively. Dotted line indicates threshold of detection.

surface receptors promoting cellular functions [12]. Therefore, we investigated cellular pathways involved in the induction of humoral responses by skin LC, by isolating human primary skin LC at steady state or after 3 days of migration *in vitro* (as depicted in Fig 1A). Strikingly, while skin LC were maturing, expression of ICOSL, OX40L and BAFF was upregulated, confirming that skin LC exhibit pro-Tfh functional makers during their migration (**S7A and S7B Fig**). Of note, the same gene modulation was observed when maturing CD34-LC (**S8 Fig**). We then compared the transcriptional profile changes in maturing skin LC and investigated messengers for pro Tfh/B cell functions. Principal component analysis (PCA) distinctly showed two different clusters (**S7C Fig**), underscoring a particular transcriptomic signature after migration, as already mentioned in another previous studies [20]. Gene expression analysis showed 5813 genes (3792 upregulated, 2021 downregulated) that were differentially expressed in migrating LC. We highlighted 39 genes involved in different pathways of interest (**S7D Fig**). Strikingly, genes involved in T cell co-stimulatory pathways (*TNFSF4/OX40L, ICOSLG, CD274/PD-L1*), or genes potentially linked to the humoral immune response induction (*TNFSF13B/BAFF*) were upregulated. A second unsupervised RNAseq analysis revealed a specific transcriptional program induced in skin LC by the targeting antibody, although no canonical pathway could be established from this set of genes. Though the mechanism by which anti-langerin vaccine stimulates skin LC to produce pro-Tfh/B cells cytokines remains elusive, our results still hightlight the inductioin by the targeting antibody of a specific transcriptional program induced in skin LC that potentially supports the Tfh/B cells functions after their migration. We thus tested *in vivo* whereas αLC.Env vaccine induced Env-specific B cells responses in a Tfh-dependent manner.

## Targeting HIV-1 Env *in vivo* to LC enhances antigen-specific B cell responses

In mouse models, monoclonal antibodies targeting the Langerin receptor drive Tfh and humoral responses against either model or Flu antigens, or the IgG4 backbone of the antibody [16, 17]. We used transgenic mice expressing human Langerin (huLangerin-DTR mice) specifically on LC [21] and compared HIV-1 Env-specific B cell responses after intraperitoneal (ip.) immunization with 1μg of αLC.Env or a non-targeting IgG4 control mAb fused with Env (IgG4.Env), respectively. Anti-Env antibodies were detectable after vaccination with αLC.Env and increased slightly after a boost at day 23 (0.6 ± 0.2 ng/mL), whereas no significant level was detectable when immunizing with IgG4.Env (**S9A Fig**). Yet, most of the humoral response was certainly masked by the human IgG4 backbone of our vaccine (**S9B Fig**). Therefore, we replaced the human IgG4 backbone of the vaccine by a mouse IgG2b, limiting thus the anti-vaccine responses generated in the mouse studies. We also used a mAb, which cross-react with the mouse Langerin, as previously reported by our team [15]. The gp140z antigen was non covalently associated to the anti-Langerin or non-targeting vehicle, using cohesin-dockerin bacterial domains for their association, as previously described [15]. Thus, the same produced antigen was used in animals vaccinated with the anti-Langerin targeting mAb (αLC.Env) or non-targeting mAb (mIgG2b.Env). To highlight the immune functions promoted by epidermal LC during immunization, we decided to administer the vaccine to the *Xcr*1$^{DTA}$ mice in which Langerin expressing dermal DC (cDC1) are depleted [22], and second to use the subcutaneous route (sc) of administration. Levels of Env-specific IgG in sera were measured by a

Luminex-beads assay [23]. Strikingly, we observed more than one log difference of anti-Env IgG in mice immunized with αLC.Env vaccine as compared to the mIgG2b vaccine (**Fig 5A**). To further investigate the activation of the Tfh/B cells responses induced by the vaccine, we looked for Tfh cells within the draining lymph nodes (dLN) and spleen (**S10A Fig**). We show that absolute numbers of Bcl6$^+$ Tfh cells was significantly increased in the dLN of both $Xcr1^{DTA}$ mice and littermate controls ($Xcr1^{Cre-mTFP1}$) vaccinated by the αLC.Env (**Fig 5B**). Among splenocytes of $Xcr1^{DTA}$ mice, the absolute numbers of Bcl6$^+$ Tfh cells was significantly higher with αLC.Env than with the non-targeting mIgG2b.Env vaccine ($P < 0.05$). Then, we looked for B cell populations within the spleen and dLN (**S10B Fig**). We show a significant expansion of a GC B population (GL7$^+$ FAS$^+$) in both organs of αLC.Env vaccinated $Xcr1^{DTA-}$ mice, as compared to $Xcr1^{DTA}$ animals vaccinated with the mIgG2b vehicle ($P < 0.05$ and $P < 0.001$ in dLN and spleen, respectively) (**Fig 5C**). Further investigations in spleen of $Xcr1^{DTA}$ mice highlighted a higher amount of plasma cells (defined as CD138$^+$ IgD$^-$) and anti-body-secreting plasma cells (defined as CD138$^+$ IgD$^-$ IgM$^+$) in animals vaccinated with the targeting vector (**Fig 5D and 5E**). Absolute numbers of memory B cells (defined as CD38$^+$ IgD$^-$) were significantly higher in $Xcr1^{DTA}$ mice vaccinated with the αLC.Env vaccine ($P < 0.05$). Finally, we searched for Env-specific B cells in spleen using a trimeric fluorescent HIV-1 Env as previously done in the lab (**Figs 5G and S11**) [3]. Eighty percent of $Xcr1^{DTA}$ animals vaccinated by the αLC.Env showed the presence of Env-specific GC B splenocytes, which was significant to the non-vaccinated WT mice ($P < 0.01$). Responses were higher than to the mIgG2b.Env group but still non-significant. Taken together, we showed a significant advantage of our anti-LC HIV-1 vaccine to promote GC-Tfh responses and Env-specific humoral responses.

## Discussion

We report here that targeting the HIV-1 Envelope to human Langerin promotes antigen-specific humoral responses both *in vitro* and in a mouse model. Human LC induced Tfh cells with the secretion of functional cytokines (IL-21). The αLC.Env mAb promotes activation of autologous naïve B cells with the production of IgG, but also the secretion of Env-specific Ig by allogenic B cells from HIV-1$^+$ individuals. Finally, using primary skin LC we provided mechanistic evidence of the pro-Tfh/B cell stimulating functions *via* expression of co-stimulation molecules such as ICOSL, OX40-L, APRIL and BAFF. We showed *in vivo* that Tfh cells detected in dLN or spleen of mice depleted for cDC1 ($Xcr1^{DTA}$) have a GC profile (Bcl-6$^+$) after vaccination with the αLC.Env mAb. Moreover, targeting HIV-1 Envelop to LC in this mice phenotype resulted in increased absolute numbers of GC B cells, specific to the Env antigen, plasma cells with a biased profile of Ab secreting and memory B cells. In addition, we detected higher levels of Env-specific IgG in the serum of the αLC.Env-vaccinated $Xcr1^{DTA}$ mice, than in mice immunized with the non-targeted antigen.

Previous studies generated in animal models and in *ex vivo* cultures of human PBMCs demonstrated that targeting antigen to human DC is a potent tool to elicit robust vaccine responses [3, 5, 7, 17, 19, 24–27]. Today, it is established that skin dendritic cells are heterogenous, can be distinguishable by surface markers and have distinct functional specializations [12, 13, 20]. By instance, CD33$^{low}$ populations are transcriptionally similar to LC. They do not function as APC and are not targeted by HIV, whereas CD11c$^+$ epidermal DC can transmit the virus [28]. Enzymatic digestions and procedures for isolating skin immune cells impacts on the detection of surface markers and thus on each DC subpopulation (e.g. CD11c, CD1c) [28, 29]. Langerin-expressing DC are found in human tissues and are distinct from LC [30]. These cells also present in mucosa are prone to HIV-1 and to transfer the virus to the CD4$^+$ T cells populations

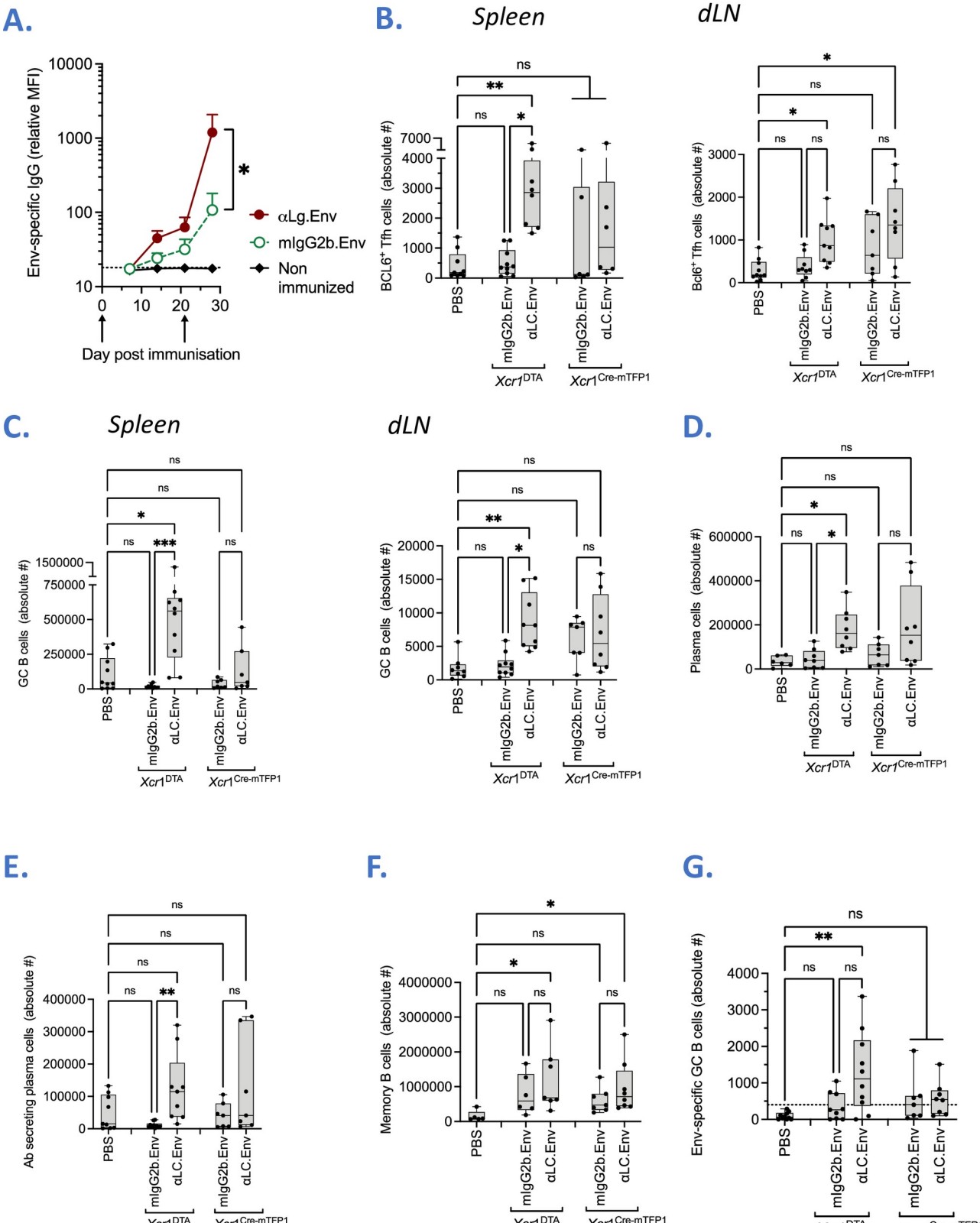

**Fig 5. The *in vivo* targeting of HIV-1 Env to the skin LC promotes Tfh responses, GC B cell responses and antigen-specific humoral responses.** (**A**) Serum antibody responses to the Env antigen. *Xcr1*[DTA] mice were immunized sc with 1 µg of αLC.Env (plain circles) or 1 µg of mIgG2b.Env (open

circles). Non-immunized $Xcr1^{DTA}$ mice were used as controls (squares). Antibody titers were measured by Luminex assay, using trimeric gp140z Env-coated beads. Means values of 6 mice per group (± SEM) are indicated. Dotted line: background of fluorescence evaluated in naïve mice. MFI, mean of fluorescence intensity. **(B)** Absolute number of Bcl6$^+$ Tfh cells in dLN (left) and spleen (right), one week post boost (day 28 post prime), in $Xcr1^{DTA}$ versus $Xcr1^{Cre-mTFP1}$ mice. B6 mice were used as controls (namely PBS). **(C)** Same as in (B), measuring absolute number of GC B cells in dLN (left) and spleen (right). **(D-F)** Same as in (B) measuring absolute number of memory B cells, plasma cells and antibody-secreting plasma cells in spleen. **(G)** From IgM$^-$ GC B cells in the spleen, absolute numbers of those specific to the Env gp140z antigen are shown. Dotted line indicates threshold of positivity (mean of B6 values + 3 SD). Non-parametric Kruskal-Wallis tests were performed (*; $P < 0.05$; **, $P < 0.01$; ***, $P < 0.001$; ns, non-significant).

[31]. Also, the cDC1 subset in mouse dermis express Langerin, and presents equivalent functions with dermic human dendritic cells subsets [32]. Therefore, one cannot exclude *in vivo* that Langerin$^+$ cDC could capture the αLC.Env vaccine or that antigen captured by LC could be transferred to another DC subset, as observed for HIV-1 particles [33]. Testing the targeting antigens to Langerin would require a functional comparison of other DC subsets present in the epidermis (as well as in the dermis), besides "truly" LC, and would constitute major steps for a better knowledge of the skin immunobiology and efficient development of vaccine. Our *in vivo* studies point to the generation of a more specific GC reaction in the SLO of mice when LC are targeted, especially in the absence of cDC1 ($Xcr1^{DTA}$) suggesting that LC might be the most prone subset to sharpen the specific humoral immune response after vaccination. On the other side, we have presently evidence that targeting HIV-1 Env to the human Langerin receptor potentializes antigen-specific B cells responses *in vitro*.

The direct transport of vaccine antigen to the dLN after targeting skin LC was visualized in NHP immunized with αLC.Gag [34] and in mice in which LC migrate to lymph node B cell areas where they drive the expansion of GC B cells [17]. Targeting lectin-like receptor LOX-1 expressed by specific DC subsets requires Poly:ICLC as adjuvant during immunization [19]. Interestingly, humoral responses were induced here in the absence of adjuvant, as reported with anti-Langerin mAbs [16, 17], mAbs against another receptors such as Clec9A [35] or by targeting XCR1 *via* the interaction of the XCL1 ligand [27]. Whilst HIV-1 infection has been shown to induce the maturation LC [36], maturation markers were not significantly upregulated after targeting the Langerin receptor with the αLC.Env vaccine. In fact, human LC drive the differentiation of autologous Tfh cells with or without prior targeting of antigen, demonstrating that human LC exhibit an intrinsic potency to drive Tfh cells, the latter being a prerequisite to help LC to present antigen to B cells and thus induce functional responses (class-switch and IgG production). However, LC potency to elicit Tfh and B cell responses was significantly intensified by targeting Langerin and not another receptor (CD40). By targeting LC, it appears that this impacts the cytokine environment during the LC/T/B 3-cell conjugate, allowing Tfh-licensed LC to deliver antigen to the B cells. The exact mechanism by which targeted LC induce Tfh cells and associated B cell functions remains to be determined. TGFβ acts together with IL-12 or IL-23 to induce the expression of various Tfh cell molecules by human naïve helper T cells [37]. Our results did not reveal that IL-12 or IL-23 might be important cofactors for the differentiation of functional Tfh cells by human LC. IL-13 and GM-CSF were highly secreted in skin LC/Tfh/B cell cultures, and the production of CXCL13 was increased by αLC.Env. CXCL13 levels have been associated with the breadth of the neutralizing antibody response in HIV-1 elite controllers, emphasizing the role of this chemokine in antibody maturation [38]. CXCL13 and GM-CSF have been shown to increase humoral and GC-B responses after vaccination [39]. IL-13 secreting CD4$^+$ T cells were also correlated with improved control of HIV-1 viremia after vaccination and interruption of antiviral treatment [4]. Importantly, BAFF and APRIL activate a pathway involved in the stimulation of B- and T cell function and their regulation of humoral responses [19]. The production of BAFF by LC during their maturation was established by transcriptomic analysis (see S7 Fig). Our data also denoted APRIL

was produced in skin LC (or CD34-LC) cocultured with T- and B cells. Therefore, a cytokine environment suitable for the B cell responses is launched. The presence of Th1 or Th2 cytokines implies a specialized support of B cells by the differentiated Tfh cells, as stated with Tfh cells expressing CXCR3 and/or CCR6 [40]. On the other hand, the production of immunoregulatory cytokines like IL-10 suggests conceivable regulatory mechanisms, that could be driven by Tfr cells [41].

Using a complete naïve autologous model, we show that the targeting of LC can drive the activation of naïve B cells as well as their isotype switching. When culturing total B cell from allogeneic HIV-1[+] patients, we still observe a positive impact of the targeting on the production of Env-specific Ig by memory B cells. The low amount of secreted IgG in these coculture systems performed without adding any cocktail of cytokines or costimulatory molecules (e.g. CD40L) precluded to analyze functional capacities of these antibodies beyond HIV Env binding. A potential interest of LC targeting is to carry directly HIV-1 Env antigen to the GC and thus favoring hypermutation cycles of B cells which has been shown to be requested the generation of bNAb against HIV-1. Whether anti-LC targeting could be a useful vehicle for sequential immunization with Env sequence from the transmitted founder virus and mutant clones [42], or for delivery of specific VRC01 targeting germline immunogen is under evaluation.

Besides, the prevention of the HIV-1 infection at the mucosal portal of entry may be facilitated by optimizing formulations for intravaginal immunization. In distinctive vaccine models, it has been reported that intravaginal application of antigens induced either antigen-specific IgA and IgG antibody secreting cells, or CD8[+] T cell responses in the female genital tract [43, 44]. We had the opportunity to control the binding of our αLC.Env candidate vaccine to the human vaginal LC ([45], and see **S12 Fig**). Therefore, the targeting of vaginal LC using anti-Langerin vehicle could be an optimal delivery modality to induce systemic and local immune antibody responses.

Advances of the last decade in the knowledge of DC biology paves the way for the development of innovative HIV-1 vaccines. Targeting human LC is a promising approach since of the intrinsic property of LC to drive the differentiation of Tfh cells and their aptitude to cargo vaccine antigen directly to the dLN B cell zones where humoral responses are initiated. Therefore, targeting human LC arises as an innovative efficient way to induce HIV-1 protective responses that should be tested in future clinical trials.

## Materials and methods

### Ethics statement

Cord blood samples were obtained from the CRB-BSC (Paris, France). Whole blood samples were collected from HIV-cART treated patients followed in Henri Mondor Hospital (Créteil, France). The study protocol was approved by the regional investigational review board (CPP Île-de-France VII and IX) with approval reference 10–023. Fresh skin samples were obtained from healthy volunteers undergoing plastic surgery of abdomen. All the subjects provided informed written consent to participate in the study. The study protocol was performed in compliance with the tenets of the Declaration of Helsinki. The vaginal tissue donors in this study provided their written informed consent. The Comité de Protection des Personnes (n° 2014/42NICB), the Assistance Publique des Hôpitaux de Paris (n° VAL/2016/2011.277/01), the clinical research committee of the Institut Pasteur, Paris, France (n° 2013.23) and the Comité Nationale de l'Informatique et des Libertés (n° 911472 v3) approved the study.

### Mice and immunization schemes

*Xcr1*[Cre-mTFP1] and *Rosa26*[LSL-DTA] (B6.129P2-Gt(ROSA)26Sor[tm1(DTA)Lky/J]),mice were described previously [46, 47]. *Xcr1*[DTA] were generated by crossing *Xcr1*[Cre-mTFP1] mice to

*Rosa26*[lsl-DTA] mice in which Cre-mediated excision of a *loxP*-flanked transcriptional STOP element triggers expression of diphtheria toxin fragment A (DTA), and results in the constitutive ablation of cDC1 [46]. Mice were kept on C57BL/6 (B6) background and housed with food and water *ad libitum*. *In vivo* procedures followed protocols approved by the Ethics Committee of Marseille in accordance with institutional, national, and European directives for animal care (approval APAFIS#28232–2021020512136337). Eight to eleven weeks old, male and female *Xcr1*[DTA] mice and littermates (*Xcr1*[Cre-mTFP1]) were subcutaneously (sc) immunized without any adjuvant in the axillary flank in a prime-boost scheme applied with 21 days of interval. Doses corresponding to 1µg of Env bound to an anti-Langerin construct (αLang.Env) or in a non-targeting form (mIgG2b.Env) were applied sc in the axillary flank of mice were randomly distributed between groups. WT B6 (purchased from Janvier) and *Xcr1*[DTA] sc PBS applied mice were used as control groups in **Fig 5A and 5B–5G**, respectively. Blood was collected each week after prime until the euthanasia, at day 28, and analyzed for anti-Env IgG levels in serum, by Luminex. Spleen and draining lymph nodes (brachial and axillary) were collected and cells were immunophenotyped by FACS. Gating strategies for T and B cells are schematized in **S10 Fig**.

## Recombinant proteins

The anti-human Langerin mAb recombinant human IgG4 antibodies used variable regions derived from an in-house anti-human langerin hybridoma (15B10, Genbank accession numbers for H chain KF021226 and L chain KF021227). This antibody is specific for the ectodomain of human Langerin and it does not cross-react with mouse Langerin. The HIV-1 Envelop Clade C 96ZM651 gp140 sequence was inserted at the IgG4 heavy-chain C-terminal codon distal of an *in-house* anti-Langerin mAb (clone 15B10 [18]) *via* a flexV1 flexible linker spacer [48]. Control Abs were derived from a non-targeting IgG4 [15], an anti-CD40 (αCD40) [9] and an anti-DCIR (αDCIR) [49] mAb and were fused with HIV-1 Env or HIV-Gag [18], as described above. HIV-1 Env gp140z with a trimerization domain were produced *in-house* [50]. Methods for cloning vectors, deriving stably transfected Expi-CHO-S cells, capturing on protein A and purifying of secreted recombinant antibodies by FPLC (Akta, GE Healthcare) have been described previously [7]. All recombinant mAbs were stored in 125 mM cavitron cyclodextrin buffer (Ashland, US) (4 mg/mL at 4˚C). Quality assurance tests were done as follow: i) SDS-PAGE followed by Coomassie staining, ii) endotoxin level below 0.5 ng/mg of protein and iii) FACS-binding assay on LC (see **S1 Fig**). His-tagged trimeric HIV-1 Envelop proteins were produced in stably transfected Expi-CHO-S cells, purified on HIS-capture column (ThermoFisher) and then biotinylated according to manufacturer's instructions (Avidity, US). The 4C7 anti-langerin or non-targeting mAbs with a mouse IgG2 backbone and associated with the dockerin (doc) domain were produced *in-house*, as well as cohesin(coh)-associated gp140z proteins [15]. Associations were performed by an equimolar mix of Coh antigen and doc vehicle, 1h at 37˚C in PBS containing calcium and magnesium.

## Cells

The Expi CHO-S cell line were obtained from Thermo Fisher Scientific (MA, USA). Cells were cultured in a medium composed of half CD-CHO Medium (Thermo Fisher) and half CHO medium 5 (Sigma Aldrich), supplemented with Glutamax (4mM, Thermo Fisher), primocin (1 mg/mL), and puromycin (10 µg/mL) (Invivogen, France). *In vitro*-generated LC, DC, B and T cells were cultured in RPMI with 10% fetal bovine serum (FBS) (Biowest, France) (R10), while LC isolated from skin explant were cultured in Iscove/RPMI medium with 10% FBS (I10). All media were supplemented with 2 mM L-glutamine, 100 U/mL penicillin and 100 ug/mL streptomycin (Thermo Fisher).

## Cell isolations from human skin explants

Isolation of skin epidermal LC was performed as previously published [33]. The layer of fats were removed and skin samples were incubated 2 hours in PBS with 20 μg/mL of Gentamycin and 0.5 μg/mL of Amphotericin B (Life Technologies) at 4˚C. Split-skin grafts of 0.3 mm were harvested from skin tissue using a dermatome (Zimmer Biomet, France) and placed in I10, completed with 1% of non-essential amino acids, 1% of sodium pyruvate (Gibco), 20 μg/mL Gentamycin and 0.5 μg/mL Amphotericin B, and treated with dispase II (0.5 U/ml, Life Technologies) overnight at 4˚C to separate dermis from epidermis. Epidermis were either incubated in I10 for 3 days at 37˚C to let LC migrating out of the epithelium (migrated LC), or immediately digested with Trypsin (0.05%, Gibco) with DNAse I (5 μg/mL, Roche) for 20 minutes at 37˚C. Bulk epidermal cells were stained, LC were then sorted by flow cytometry using CD45, HLA-DR, CD1c, and CD1a expressions and used immediately (steady state LC). Cell phenotype before and after flow cytometry sorting is illustrated in **S1 Fig**.

## *In vitro* differentiation of human LC

Mononuclear cells from blood and cord blood samples were Ficoll-isolated (Alere Technologies). The differentiation of hematopoietic stem cells from cord blood into LC (named CD34-LC) was performed as previously published [17]. $CD34^+$ hematopoietic progenitors from the cord blood were isolated using CD34 magnetic beads (Miltenyi Biotec) (purity >95%). Cells were sorted with an autoMACS Pro Separator and were cultured for 10 days with two sets of cytokines: SCF (25 ng/mL), TNFα (2.5 ng/mL), and GM-CSF (20 ng/mL) (Miltenyi) at day 0, followed by addition of TGFβ, 6 ng/mL, TNFα, 2.5 ng/mL, and GM-CSF, 20 ng/mL at day 4. The CD34neg autologous cell fraction was kept frozen until obtention of *in vitro* differentiated LC (CD34-LC). CD34-LC were harvested, controlled for their phenotype and sorted using CD1a magnetic beads (Miltenyi Biotec).

## Cell culture of CD34-LC

CD34-LC were treated with αLC.Env or control mAbs (10 nM) 1h at 37˚C in R10 and washed. Autologous naïve $CD4^+$ T cells were isolated from the $CD34^{neg}$ fraction of the same donor using a microbead isolation kit (Miltenyi Biotec). The purity of sorted naïve $CD4^+$ T cells was controlled by FACS (see **S2 Fig**). Then, naïve $CD4^+$ T cells were cultured without any cytokine and added to CD34-LC (ratio $5x10^4$ CD34-LC: $2x10^5$ T cells). Anti-CD3/CD28 beads (ThermoFisher) were added in each well at day 2 of culture to increase T cell survival and differentiation. T cell differentiation profile and proliferation were analyzed at day 4 using flow cytometry. Then, depending on the experiment, autologous naïve B cells or allogenic $HIV^+$ total B cells were sorted using magnetic beads, and $10^5$ B cells were added in each well. Naïve B cells were primed following a method previously described [19]. Cells were lifted and culture supernatants were collected at day 6 or 13 after adding B cells. B cell phenotype was assessed at day 6. Ig production was analyzed by ELISA and Luminex assay using culture supernatants collected at different times.

## *In vitro* differentiation of MoDC

$CD14^+$ monocytes were isolated using a microbead isolation kit (Miltenyi Biotec) and differentiated into MoDC as previously published [51]. Purified monocyte populations were isolated with $CD14^+$ magnetic beads and cultured with 20 ng/ml GM-CSF, and 2 ng/ml IL-4 (Miltenyi Biotec) for 5 days with a boost at day 2. Phenotype of immature DC was controlled by FACS as $CD11c^+$, $CD1a^+$, $HLA-DR^+$, $HLA-I^{low}$, $CD86^{low}$, $CD207^-$.

## Antibodies

For flow cytometry on human cells, fluorochrome-conjugated antibodies to CD45 (2D1), HLA-DR (G46-6), CD86 (2331 (FUN-1)), CD3 (SK7), CD4 (RPA-T4), PD-1 (EH12.1), CXCR5 (RF8B2), Bcl-6 (K112-91), CD45RO (UCHL1), CD45RA (HI100), FoxP3 (259D/C7), CD25 (2A3), CD19 (HIB19), CD21 (B-ly4), CD27 (M-T271), IgD (IA6-2) were provided by Beckman Dickinson (BD Biosciences, Franklin lakes, NJ). Anti-CD1c (L161), anti-CD1a (HI149), anti-MHC-I (W6/32) and anti-CD207 (10E2) were provided by Biolegend; San Diego, CA. Anti-IgA (IS11-8E10) and anti-CD138 (44F9) were provided by Miltenyi Biotech. Anti-ICOS (ISA-3) was provided by eBioscience. For flow cytometry on mouse cells, single cell suspension of skin-draining LNs were incubated *ex vivo* with AF-647-conjugated huIgG4 [17] and a cocktail of fluochrome-conjugated antibodies to CD19 (1D3), anti-CD38 (90) (Thermo Fisher), IgM (RMM-1), GL7 (GL7) (Biolegend) and B220 (RA3-6B2) (Tonbo Biosciences).

Splenocytes and single cell suspension of axillary and brachial draining lymph nodes of mice were stained for flow cytometry. Freshly harvested cells were incubated with the following fluorochrome-conjugated mouse antibodies: anti-Bcl6 (K112-91), anti-CD4 (GK1.5), anti-CD5 (53–7.3), anti-CD8a (53–6.7), anti-CD11b (M1/7), anti-CD19 (1D3), anti-CD25 (PC61), anti-CD38 (90/CD8), anti-CD44 (IM7), anti-CD45R (RA3-6B2), anti-CD62L (MEL-14), anti-CD95 (JO2), anti-CD134 (OX86), anti-CD138 (281.2), anti-CD154 (MR1), anti-CD161c (PK136), anti-CD183 (2G8), anti-CD278 (15F9), anti-CD279 (29F.1A12), anti-Fc (2.4G2), anti-Foxp3 (FJK-16S), anti-GL7 (2G8), anti-IgD (11-26C-2a), anti-IgG (Poly4053), anti-IgM (FJK-16), anti-MHCII (M5/114.15.2); obtained from BD Biosciences. To assess gp140z-specific memory B cells, biotinylated gp140z protein was pre-incubated with cell suspension and revealed by BV650-conjugated streptavidin mixed to the cocktail of fluorochrome-conjugated surface antibodies, as previously described [3].

## Flow cytometry

All staining was performed at 4˚C for 30 minutes. LIVE/DEAD near-IR staining kits and fixable aqua (Life technologies) or Zombie UV (BioLegend, CA, USA) were used to discriminate live and dead cells. All the flow cytometric plots presented in this article were pre-gated on live (using Live/Dead stain) and singlet events. Proliferation analysis was performed using CellTrace Violet Proliferation Kit (Molecular Probes). For intracellular staining, cells were fixed, permeabilized and stained following manufactor's instructions of kit eBioscience Foxp3 / Transcription Factor Staining Buffer Set (Invitrogen, CA, USA). Cell acquisition was performed using BD FACSymphony and LSR II flow cytometer (BD), cell sorting was performed using an Influx flow cytometer (BD). Data were analyzed with FlowJo software (TreeStar, Ashland, OR).

## ELISA

Samples were incubated 1h at RT on plates coated with an anti-human IgG polyclonal antibody (Jackson ImmunoResearch), using a serial-diluted reference 10–1074 mAb (NIH AIDS reagent program, #12477) for quantitative analysis. An anti-human IgG polyclonal antibody coupled with HRP (Jackson ImmunoResearch) was used as secondary antibody. Absorbance of each well was measured at 450 nm (reference 405 nm) using a Tristar2 reader (Berthold Technologies).

For the detection of Env-specific Ig, plates were coated with 1μg of an anti-human total Ig (ThermoFisher). Diluted samples were added for 8h at RT, before adding biotinylated (Avidity) HIV-1.Env trimer overnight at 4˚C (2 μg/mL). Plates were then incubated 2h at RT with streptavidin-HRP (1:40,000; ThermoFisher), followed by addition of the TMB substrate.

The detection of latent/versus active TGFβ in culture was performed by ELISA following manufacturer's instruction (R&D systems).

## Multiplex bead-based technology

T-helper cytokine production and antibody isotyping were assessed on culture supernatants collected at day 4 and day 10 respectively using multiplex bead-based technology (Luminex) kits (Procarta). Experiments were performed following provider instructions and using Bio-Plex 200 instrument (Bio-Rad Laboratories).

## Luminex beads-assay for detection of gp140z-specific IgG

We adapted the Luminex assay developed by C. Fenwick and coll. to detect anti-HIV-1 gp140z Env-specific IgG [23]. The *in-house* produced HIV-1 Env Gp140z trimer was associated with MagPlex beads using the manufacture's protocol (Bio-Rad, France). Activated beads were washed in PBS followed by the addition of 6.3 μg of protein antigen (4°C overnight under agitation). Beads were then washed with PBS, resuspended in blocking buffer and finally in 150μl of storage buffer. Beads were counted with an Auto-2000 cell counter (Nexcelom) and kept protected from light at 4°C. Luminex beads were diluted at 50,000 beads/mL in PBS and 50μl was added in a Bio-Plex Pro 96-well Flat Bottom Plates (Bio-Rad). Following two 0.05% tween PBS on a magnetic plate washer (Bio-Rad), 50 μl of individual serum samples diluted at 1/100 in PBS, were added. Binding was performed at RT for 30 min under agitation, before adding an anti-mouse IgG-PE secondary antibody (45min at 0.5 μg/mL, ThermoFisher). Beads resuspended in 80 μl of Sheath fluid (Bio-Rad) were agitated 5 min at 700 rpm on the plate shaker then read directly on a Bioplex-200 plate reader (Bio-Rad) with 50μl of acquisition volume and DD gate 5,000–25,000 settings. Median fluorescence intensities (MFI) were exported using Bioplex Manager 6.1 software.

## Statistical analysis

For statistical analyses of immune responses of mice, of the flow cytometry data or RT-qPCR, non-parametric Kruskal-Wallis tests with Dunns multiple comparison posttest were used. For in vitro phenotypical analysis of B cells at different time points, two-tailed paired T-test were performed. For statistical analyses of antibody concentration in culture supernatants, two-way ANOVA with Dunnett's correction were used (ns, non-significant; * $P < 0.05$; ** $P < 0.01$; *** $P < 0.001$). Error bars depict mean +/- standard error of the mean (SEM) in all bar graphs shown. Whiskers define the minimum and maximum of the data presented, with medians shown as bars. The GraphPad Prism statistical analysis program (Graph-Pad Prism Software, version 9) was used throughout.

## Supporting information

**S1 Fig. FACS-isolated skin LC phenotyping. (A)** Cell suspension from trypsin-digested human epidermis was stained and analyzed by flow cytometry. CD45+ HLA-DR+ populations showed 3 subpopulations, depending on the expression of CD207 and CD1a, namely populations A, B and C. Population "A" was CD11c^hi, "B" CD11c^low and HLA-DR^low, whereas "C" was CD11c^low and HLA-DR+. These skin epidermal populations was phenotypically close to epidermal myeloid populations already described [28]. **(B)** Skin epidermal cell suspension was sorting by the Influx cell sorter, using Live dead, CD45, HLA-DR, CD1a and CD1c markers. **(C)** Isolated steady-state skin cells were controlled by FACS. They show typical markers of LC, meaning CD207+, CD1a^hi, CD1c+, CD11c^low, and were not matured (MHC class I and CD86

low expression). **(D)** As comparison, cells migrated in culture from skin explants were phenotypically activated (MHC class I$^{hi}$, CD86$^{hi}$) with expression of LC markers but at a lower level (CD1a$^{int}$ CD207$^{int}$).
(TIF)

**S2 Fig. Phenotyping of sorted naïve CD4$^+$ T cells.** CD4$^+$ naïve T cells from cord blood were controlled by FACS for the absence of Tfh cells, based on the expression of PD-1, CXCR5 and Bcl-6.
(TIF)

**S3 Fig. Capture of αLC.Env and control mAbs by skin LCs and CD34-LCs. (A)** Binding assay of αLC.Env mAb on epidermal LCs by immunofluorescence assay. Sections of frozen skin explants were treated with αLC.Env or IgG4.Env mAbs. Binding to skin LCs was controlled using a commercial anti-CD207 mAb (12D6). Pictures are representative of 4 different donors (magnification x20). Skin LCs are located within the epidermis and capture αLC.Env mAbs but not IgG4.Env. **(B)** Total epidermal cells were stained with phenotypical markers and incubated with serial dilution of αLC.Env (full red circle) or IgG4.Env (open circle). A specific binding of CD207$^+$ cells was observed for αLC.Env in a dose-dependent manner by FACS. **(C)** Representative phenotypic analysis of steady state skin LCs and unmatured CD34-LCs at the end of the differentiation process and before CD1a sorting. Cells were characterized by high expression of CD1a and CD207. **(D)** As in A, binding assay of aLC.Env versus IgG4.Env on CD34-LC. Binding was comparable in freshly isolated skin LC and CD34-LC (IC$_{50}$ = 3 nM and 1.4 nM, respectively). The binding of the αCD40.Env was tested in parallel (black triangles). **(E)** Internalization of the Langerin receptor. CD34-LC were incubated overnight (37˚C) with αLC.Env, αLC.Empty, control mAbs (αCD40.Env, IgG4.Env) or cyclo-dextrin buffer (untx, untreated). We observed a significant internalization of Langerin 16h after treatment with αLC mAbs compared to control mAbs ($P < 0.001$), suggesting a rapid endocytosis of the antigen targeted to the Langerin receptor. Data are means (± SEM) of 4 donors. Two-way ANOVA with Dunnett's correction were used for statistical analysis (\*\*\*, $P < 0.001$).
(TIF)

**S4 Fig. Differentiation of autologous CD4$^+$ T cells towards the Tfh lineage by CD34-LC. (A)** Schematics of the procedure of the CD34-LC treated with αLC.Env or control mAbs and cultured with autologous naïve CD4$^+$ T cells. **(B)** (left) Representative CD4$^+$ T cell dot plots of one out of 6 donors. CXCR5$^+$ PD-1$^+$ population (red) was Bcl6$^+$ and was proliferating as indicated by the loss of cell trace marker. (right) CXCR5$^+$PD-1$^+$ were either CD25$^+$ FoxP3$^{hi}$ (Tfr cells) or FoxP3$^-$ (Tfh cells). **(C)** Percentages of CXCR5$^+$PD-1$^+$ among CD4$^+$ T cells. **(D)** Percentages of Tfh vs Tfr cells among CXCR5$^+$ PD-1$^+$ cells after treating CD34-LC. **(E)** Replication indexes of Tfh cells and Tfr cells were calculated from the loss of cell trace marker.
(TIF)

**S5 Fig. IgG production by total B cells was specific to the targeting of the langerin receptor expressed by CD34-LC. (A)** Vaccine dose-dependent B cells responses of αLC.Env treated CD34-LC (plain circles) and IgG4.Env (open circles). **(B)** IgG production was compared to αLangerin mAbs fused with HIV-1 Gag antigen, or αDCIR, αCD40 and non-targeting IgG4 vehicles associated with the HIV-1 Env gp140z antigen. **(C)** IgG4 detected in cell co-cultures was not derived from the treatment of CD34-LC with IgG4-based vaccine. CD34-LC were treated with αLC.Env or control Abs and cultured 10 days without adding T- or B-cells (n = 3). As controls, CD34-LC were treated and cultured as in Fig 2 (n = 3). Concentrations of IgG4 in culture supernatant was measured by ELISA, using an anti-human IgG4 mAb (ThermoFisher) for coating. Non-parametric Kruskal-Wallis tests were performed (\*; $P < 0.05$; \*\*,

$P < 0.01$; ***, $P < 0.001$; ns; non-significant).
(TIF)

**S6 Fig. Targeting CD34-LC promotes a specific cytokine environment. (A)** As in Fig 3, multiplex beads analysis and comparison of culture supernatants at day 4 or 10 (as depicted in Fig 2A) with CD34-LC treated by αLC.Env, αCD40.Env or IgG4.Env, cultured alone (column 1), or with autologous naïve CD4+ T cells (column 2) for 4 days, and with or without (column 3) autologous naïve B cells (column 4) or total allogenic B cells (column 5). Color scale indicates median values of each condition (n = 4 to 9) and is expressed in pg/mL. Values of IL-21 and APRIL are detailed in **(B)**, and IL-18 and CXCL13 in **(C)**, respectively. Dotted line indicates threshold of detection.
(TIF)

**S7 Fig. Expression of pro-Tfh markers during activation of human skin LC. (A)** Expression of maturation surface markers of skin LC was monitored by flow cytometry before and after migrating out of the epidermis (n = 4) **(B)** Real-time qPCR of skin LC transcripts. Total mRNAs from steady state epidermal cells vs migrated skin LC (n = 5) were isolated. Relative level expressions of transcripts associated with LC activation (CD207, TGFβ), T cell co-stimulation (ICOSL, OX40L), and gene genes potentially linked to the humoral immune response induction (BAFF) were quantified by qPCR. Statistics were obtained using the non-parametric Mann-Whitney test (*, $P < 0.05$; **, $P < 0.01$; ns, non-significant). **(C)** RNAseq analysis on primary skin LC. Epidermal LC of 6 healthy donors were isolated either at steady state or after 3 days of migration in culture. Unsupervised PCA of the transcriptional analysis revealed two distinct populations. **(D)** Modulation of the expression of genes associated with Tfh/B cell signaling pathway, antigen uptake and markers of activation are indicated on right.
(TIF)

**S8 Fig. CD34-LC express pro-Tfh markers after maturation. (A)** Schematic of the procedure for maturing CD34-LC. Once differentiated, CD34-LC were cultured 24h without any cytokines to induce their maturation. **(B)** Expression of maturation markers of CD34-LC was monitored by flow cytometry (n = 9). **(C)** Total mRNAs from CD34-LC (n = 5) were isolated and real-time RT-qPCR of LC transcripts was performed as in Fig 5. Statistics were obtained using the non-parametric Mann-Whitney test (*, $P < 0.05$; **, $P < 0.01$; ***, $P < 0.001$; ns, non-significant).
(TIF)

**S9 Fig. The *in vivo* targeting of HIV-1 Env to the human Langerin receptor promotes antigen-specific humoral responses.** HuLangerin-DTR mice were immunized IP with 1 μg of αLC.Env (plain circles) or 1 μg of IgG4.Env (open circles), or with vehicle only (cyclo-dextrin buffer). **(A)** Env-specific antibody titers were measured by ELISA at day 14, 21, 31 and 38 post-immunization, using the 10.1074 antibody as reference control. Mean (± SEM) concentrations of each group (n = 3) are indicated. **(B)** Same as A, measuring anti-hIgG4 specific antibody amounts. Statistics were obtained using the Holm-Sidak's multiple comparisons test (**, $P < 0.01$; ns, non-significant). Arrows indicate dates of injection.
(TIF)

**S10 Fig. Representative gating strategy for analyzing cellular responses in dLN and spleen of immunized *Xcr1*DTA or *Xcr1*Cre-mTFP1 mice.** Single cells suspension of mice spleen and dLN were stained. **(A)** FACS-analysis and gating strategy of Tfh cells populations (spleen). Cells were gated on singlets, live and CD5+ Lineage- subsets. CD4+ CD8- T cells were separated in CD25- and analyzed for the co-expression of CXCR5+ PD-1+, characterizing Tfh cells. Tfh

cells were checked for the expression of Bcl-6 **(B)** FACS-analysis and gating strategy of B cells populations (spleen). Debris, double and dead cells were excluded, and B cells were analyzed from CD5⁻ Lineage⁻ subset (CD11b⁻ NK1.1⁻). Cells expressing low or positive values for CD19 and B220 were gated on MHC-II⁺ and then Plasma cells were identified by high expression of CD138 and null IgD. From those cells, the ones expressing IgM were identified as antibody-secreting Plasma cells. After exclusion of Plasma cells, GC B cells were gated on FAS⁺ GL7⁺. IgM⁺ B cells were excluded and IgM⁻ GC B were analyzed for HIV-1.Env trimer specificity. Non-Plasma cells, non- GC B cells FAS⁻ GL7⁻ population was analyzed for CD38 and IgD expression. Memory B cells were defined as CD38⁺ IgD⁻.
(TIF)

**S11 Fig. Representative dot-plot analysis of Env-specific B cells in spleen of immunized *Xcr*1^DTA^ or *Xcr*1^Cre-mTFP1^ and C57BL/6J controls.** To identify GC B cells specific to HIV-1. Env protein, an *in-house* trimer of gp140z.biotin was made. Splenocytes were incubated previously with the trimer, washed and then stained for surface B-subsets specific antibodies. Debris, double and dead cells were excluded, and B cells were analyzed from CD5⁻ Lineage⁻ subset. B cells were defined as MHC-II⁺ low or positive CD19 and B220 expressing cells. After exclusion of plasma cells (CD138⁺⁺ IgD⁻), GC B cells were identified on FAS⁺ GL7⁺ gate. GC B cells expressing IgM were excluded and IgM⁻ GC B were analyzed for HIV-1.Env trimer specificity. Plots show frequency of parents for Env-specific GC B cells of one representative mouse per group.
(TIF)

**S12 Fig. αLC.Env target cells expressing Langerin in human vaginal mucosa.** Langerin expressing cells of the vaginal mucosa were stained with (+) or without (-) a commercial anti-Langerin mAb (12D6) and revealed by peroxidase (brown). Concomitantly, additional sections were treated with αLC.Env and revealed by alkaline phosphatase (red). Langerin expressing cells appear located within the epithelium and cells binding αLC.Env are distributed in the same way. Arrows show examples of stained cells. Three representative donors are depicted. (magnification x10).
(TIF)

**S1 Text. Supplementary Methods.**
(DOCX)

## Acknowledgments

*VRI Program management*: Dr Mireille CENTLIVRE and Laurent HANOT. *Technical assistance for biological samples*: the plastic surgery service of Mondor's hospital (Prof. Jean-Paul MENINGAUD), the gynecology-obstetrics service of the Kremlin Bicêtre hospital, Prof. Sophie HÜE, Dr Romain BOSC, Dr Laura FERTITTA, Marie-Thérèse NUGEYRE, Dr Valérie LAPIERRE and Dr Clément BONAMY. *Cytometry know-how*: Adeline HENRY, Odile RUCK-EBUSCH, and Aurélie WIEDEMANN. *Reagents*: NIH AIDS reagent program, Dr Hugo MOUQUET.

## Author Contributions

**Conceptualization:** Hakim Hocini, Gerard Zurawski, Yves Levy, Sylvain Cardinaud.

**Data curation:** Hakim Hocini, Sylvain Cardinaud.

**Formal analysis:** Jérôme Kervevan, Juliane S. Lanza, Véronique Godot, Sandrine Henri, Botond Z. Igyártó, Yves Levy, Sylvain Cardinaud.

**Funding acquisition:** Botond Z. Igyártó, Yves Levy, Sylvain Cardinaud.

**Investigation:** Jérôme Kervevan, Aurélie Bouteau, Juliane S. Lanza, Adele Hammoudi, Sandra Zurawski, Mathieu Surenaud, Lydie Dieudonné, Marion Bonnet, Cécile Lefebvre, Romain Marlin, Aurélie Guguin, Sylvain Cardinaud.

**Methodology:** Hakim Hocini, Véronique Godot, Sandrine Henri.

**Project administration:** Yves Levy, Sylvain Cardinaud.

**Resources:** Barbara Hersant, Oana Hermeziu, Elisabeth Menu, Christine Lacabaratz, Jean-Daniel Lelièvre, Gerard Zurawski, Botond Z. Igyártó, Yves Levy.

**Supervision:** Yves Levy, Sylvain Cardinaud.

**Validation:** Yves Levy, Sylvain Cardinaud.

**Visualization:** Yves Levy, Sylvain Cardinaud.

**Writing – original draft:** Sylvain Cardinaud.

**Writing – review & editing:** Aurélie Bouteau, Juliane S. Lanza, Sandra Zurawski, Mathieu Surenaud, Marion Bonnet, Hakim Hocini, Romain Marlin, Elisabeth Menu, Christine Lacabaratz, Gerard Zurawski, Sandrine Henri, Botond Z. Igyártó, Yves Levy, Sylvain Cardinaud.

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
