## [Decision Letter · Decision Letter 0]

13 Jan 2021

Dear Dr CARDINAUD,

Thank you very much for submitting your manuscript "Targeting human langerin promotes HIV-1 specific humoral immune responses" for consideration at PLOS Pathogens. As with all papers reviewed by the journal, your manuscript was reviewed by members of the editorial board and by several independent reviewers. In light of the reviews (below this email), we would like to invite the resubmission of a significantly-revised version that takes into account the reviewers' comments. Please not that the reviewers noted a number of points that need to be addressed including a general editing of the manuscript.

We cannot make any decision about publication until we have seen the revised manuscript and your response to the reviewers' comments. Your revised manuscript is also likely to be sent to reviewers for further evaluation.

Sincerely,

Alexandra Trkola

Associate Editor

PLOS Pathogens

Thomas Hope

Section Editor

PLOS Pathogens

Kasturi Haldar

Editor-in-Chief

PLOS Pathogens

orcid.org/0000-0001-5065-158X

Michael Malim

Editor-in-Chief

PLOS Pathogens

orcid.org/0000-0002-7699-2064

Reviewer's Responses to Questions

**Part I - Summary**

Reviewer #1: Kervevan and colleagues have fused HIV-Env to a langerin antibody (anti-Lang-env) and investigated its ability to induce an immune response via LCs. Firstly, they show in vivo (using mice expressing human langerin) that anti-Lang-env leads to robust production of IgG4 env specific antibodies and a high frequency env specific B cells that express germinal centre markers. They next targeted anti-Lang-env to human cord blood derived CD34-LCs and showed that autologous (cord blood?) naïve CD4+ T cells were differentiated into Tfh-like cells. When mixed with autologous B cells this led to the generation a very small (but significant) percentage of memory B cells some of which were of a CD21+ resting memory phenotype. Treatment of CD34-LC with also led to production of IgG2 and 4 (not IgG1 and 3) in this assay and then anti-env antibodies from B cells from allogeneic HIV= donors apparently providing evidence of a functional interaction between naïve B cells and co-cultured autologous CD34-LC. They also investigated the costimulatory receptors and the array of cytokines secreted by CD34-LCs after ant-Lang-env treatment and showed some were consistent with Tfh differentiation and B cell function. Finally, they show in primary human skin LCs that similar cytokine expression profiles were seen.

Although we already know the approach (i.e. using anti-LC antibodies) works – these authors showed this in Ref#16 – I believe with major revisions this manuscript is of interest to the readers of PLoS Pathognens.

The underlying hypothesis is that targeting HIV env to human LCs via anti-langerin antibody specifically induces Tfh cells and thence B cell memory and specific anti-env antibody secretion by antibody secreting cells.

They have used a complimentary mix of transgenic mouse, model human cord blood derived CD34-LCs and authentic human LCs and B cell from HIV+ patients (under allogeneic stimulation) to try to test this hypothesis Although these are valiant attempts the interpretation of the experiments falls short of proof of the hypothesis because of the lack of some controls, especially in the models systems.

Human CD34-LCs are very different to bona fide LCs by transcriptomics and HIV binding C-type lectin expression, overlapping with MDDCs and probably do not exist in vivo, at least not in stratified squamous epithelium. Thus, they need to show that Tfh are only induced by langerin and not other C-type lectin stimulation in these cells, confirm their findings with Tfh induction with authentic epidermal LCs and that other tissue DCs (preferably from anogenital epithelium or skin) do not have these properties. Indeed, we already know that skin DC (incl LC) drive TFH Ref#15 and GC responses Ref#12.

As env binds langerin directly then env itself (+/- conjugated to IgG4) should be compared to anti-langerin in all the above experiments to show any added benefit of anti-langerin conjugation.

The mouse immunization experiments were intraperitoneal so local draining lymph nodes would be unlikely to contain LCs requiring more distal targeting to skin draining ing lymph nodes. As they mention in discussion mucosal immunization would drain to these lymph nodes so there a comparison of the need for env to be anti-langerin conjugated becomes more cogent.

Reviewer #2: The authors report on a vaccine strategy in which antigen is fused to anti-langerin antibody. Injection of the recombinant fusion protein is thought to induce B cells to produce antibodies against the antigen. The strategy is also supposed to cause LCs to induce the differentiation of germinal center T follicular helper cells.

This is an interesting and innovative approach for introducing antigen into LCs that is worth exploring. The idea is that LCs may be better than DCs at promoting Tfh cell activation in GCs. The authors have done a considerable body of work using sophisticated cell culture methods. The experiments include relevant controls.

An overarching question about the study is whether this will be a valuable approach for an HIV vaccine. The problem with an HIV vaccine is not the inability to get an antibody response to antigen. The problem is inducing a broadly neutralizing response, or broadly acting CD8 T cells. bNAbs are produced at very low frequency in patients and only in rare individuals and no immunogens have been developed that induce a broadly neutralizing response. Targeting Gp140 to LCs is not going to solve this problem. Perhaps the approach will be useful for other specific antigens, at least if it could be shown that the immunization results in better antibodies than just injecting antigen. The study does not convincingly make that case.

A weakness with the experiments is that some of the effects are quite modest and some do not seem to be significant. Furthermore, the data do not show that the approach is any better than standard approaches of injecting purified antigen. Most of the data presented just show the antibody response to the antibody IgG4 carrier. Very little is done to look at the anti-gp140 antibody response.

An important consideration is whether the study is appropriate for PLOS Pathogens. It is not aimed at a pathogen but is an immunological report to look at Tfh differentiation and antigen targeting to LCs. It would more suited to an immunology journal, of which there are many.

Specific points

1. Fig. 1A. The figure shows the anti-IgG4 antibody response but does not show the anti-gp140 antibody response. A histogram needs to be added to show the anti-Gp140 antibody titer. The anti-IgG4 response is not what’s important, it's the anti-gp140 antibody that matters. Panel C does shows a time course of anti-Gp140, but that is not sufficient. If the approach is designed to induce antibody against the antigen linked to anti-langerin, then the titer of anti-Gp140 antibody is a central question.

2. Fig. 1B. The protocol involves adding anti-CD3/CD28 to the cocultures. The treatment is a strong T cell activator and one that would not be used in a vaccine. The vaccine needs to work in the absence of such a treatment. The authors need to demonstrate that the vaccine can stimulate a B cell response in the absence of global T cell activation.

Also, the size of the font of the text included in the flow plots needs to be increased for readability.

3. Fig. 1C. This is one of the few analyses of the anti-gp140 antibody but the data it is weak and could be interpreted to suggest that the approach is not a good one. The anti-gp140 antibody is about 1/500 of the anti-IgG4.

4. Fig. 2A. The authors switch here to human cells. They need to make this more clear by stating it at the beginning of this section in Results.

5. Fig. 2D. The effects shown are small. There is no response for Ig A, M, G, G1, G2. The IgG3 is negligible and the IgG4 is very small. The findings seem to suggest that the approach is not good.

6. Fig 3. Here the experiment uses HIV+ patient B cells. The text needs to make this more clear. The rationale is presumably to increase the number of B cell clones expressing anti-gp140 antibody. This needs to be more clearly stated. This vaccine is not a therapeutic vaccine. If it requires previously primed B cells, it is not going to work.

7. Fig. 3D. This is the only other panel that looks at the anti-gp140 response. It shows an increase in antibody to gp140 of about 2-fold over background. This seems to be weak. It does not state the titer of the anti-gp140 antibody.

8. Fig 4B. The cytokine response is difficult to judge from the figure. The legend at the right goes to >2-fold. There are only 2 cytokines that are increased (CXCL13 and IL2) and if these are less than 2-fold increases then the response is minimal. It is not possible to compare the color of the squares which appear to be gray with the colors on the legend.

9.Fig. 5. The figure is an experiment to look at the phenotype of LCs. It has nothing to do with the anti-langerin-gp140. It thus, does not fit in with the study. The authors refer to Fig. S4C, which does use the fusion protein. It would seem that this is the relevant experiment and should be in the main text, not the supplement.

Reviewer #3: In this manuscript the authors describe the use of a novel and interesting reagent i.e an Ab specific to Langerin, a molecule expressed in skin DCs to target the HIV env protein. They show that targeting this molecule to DCs in mice will lead to the development of Abs to env. They dissect mechanistically the development of these env specific Abs. They show that naive T cells added to cultures of in vitro matured Langerhans cells will differentiate into Tfh cells that will provide help in these cultures to B cells from HIV + subjects. This help will lead to production of env specific Abs. They perform RNA seq experiments on these in vitro matured Tfh cells and show that these cells adopt some of the features (cell surface markers and cytokines generic to Tfh cells).

The authors of this manuscript have developed this reagent to address an important need i.e how to enhance the production of Abs to HIV env protein. Unfortunately the manuscript falls short of providing a real solution that can be used in the context of an HIV Cure or an HIV prevention .

• What is the rationale for targeting skin DCs; while skin DCs can prime B and T cells, skin is not the most prevalent mode of transmission; hence one weakness is the relevance of this reagent

• The authors do not show that these Abs can neutralize HIV or act as broadly neutralizing Abs or prevent HIV infection in SCID hu models of HIV infection or let alone in vitro

• The experiments are descriptive, and the authors fail to provide or more importantly validate any of the suggested mechanisms of action namely promoting Tfh differentiation

• The bioinformatic analysis can be significantly improved

**Part II – Major Issues: Key Experiments Required for Acceptance**

Reviewer #1: Major points:

1. Model LCs derived from CD34+ cord blood monocytes were used. These are very different from bona fide LCs which are derived from the yolk sack. The authors should confirm with authentic tissue LCs in at least some parameters investigated in Figures 2 and 3, as mentioned above. Env and anti-langerin should be directly compared in all in vitro experiments

2. The results in supplemental Figure 4 and 5 are important and should be a main Figures.

3. The use of allogeneic HIV+ B cells in figure 3 is very artificial. Could this be augmented by testing ubiquitous antigens such as tetanus toxoid in an autologous system?

4. Why is there a difference in IgG subtype stimulation (3 and 4 vs 2 and 4) in figures 2 and 3.

5. Much of the data in Figure 5 is unremarkable. As migratory LCs undergo maturation it is really no surprise at all that they cluster differently to immature cells isolated by trypsin digestion. Significant differences in clustering between mature and immature cells has been well demonstrated (e.g. PMID 23183897). Genes associated with T cell stimulatory pathways are known to be specifically upregulated by APCs that spontaneously migrate out of dermis and epidermis as has already been well characterised in the literature (e.g. PMID: 28270408, 17082627). These and/or other similar studies should be referenced.

6. Line 258 and 259: The authors state that langerin is exclusively expressed by LCs in humans. This is not correct. Some dermal cDC2 also express langerin (PMID: 25516751) and recently anogenital langerin expressing epidermal DCs have been identified and characterised (PMID: 31227717). Indeed, epidermal DCs were shown to be far more efficient at HIV uptake and productive infection and transfer to CD4 T cells than LCs. In fact, by using CD1c to gate LCs the authors were probably selecting for epidermal DCs rather than (or certainly as well as) LCs (See PMID: 31227717 Figure 1A). The role of other human langerin expressing cells needs a detailed discussion as this envelope antibody may well be taken up by these langerin expressing cDC2 populations rather than LCs. Ideally the authors should compare epidermal CD11c+ cells to LCs.

Reviewer #2: Fig. 1A. A histogram needs to be added to show the anti-Gp140 antibody titer. The anti-IgG4 response is not what’s important, it's the anti-gp140 antibody that matters.

Reviewer #3: The authors should

• Show that these Abs can neutralize HIV infection in primary CD4 T cell models of HIV infection and in SCID hu mouse models of HIV infection

• Provide validated mechanisms that can support the hypothesis raised by the authors i.e Langerin targeted LCs can trigger the differentiation of naïve T cells to Tfh cells . In these experiments the authors should provide convincing evidence that the Tfh cells were not a contaminant of the original input . They should perform spiking experiments and titrate in Tfh as their positive control . They so do not have a benchmark to show what bonafide Tfh cells can do to enhance the production of env specific Abs.

• Adding increasing numbers of Langerin targeted LCs or of naïve cells to these cultures and show a dose dependent increase in Ab titers is an important control

• The bioinformatic analysis should involve comparisons to bonafide Tfh signatures published in the literature. The analysis while suggestive are incomplete

• The authors have identified several cytokines and cell surface markers generic to Tfh cells but they do not show that these molecules play a role in the enhancement of production of env specific Abs . They should perform blocking experiments .

**Part III – Minor Issues: Editorial and Data Presentation Modifications**

Reviewer #1: Minor points:

1. There are numerous minor grammatical errors throughout the manuscript that need addressing.

2. Line 128 – 131: Please provide a reference for this recent work (I assume ref 16).

3. Please make sure to always use the term CD34 LC (rather than LC) when referring to model CD34 monocyte derived LCs (including in all titles).

4. LC isolation. I would recommend the use of collagenase IV rather than trypsin for LC isolation as leads to the isolation of cells with intact cell surface markers and with greater viability. Indeed, trypsin has been shown to cleave CD1c which the authors use to gate on LCs (PMID: 28270408). This could also influence differences in direct env vs anti-langerin-env targeting and stimulation of LCs

Reviewer #2: some of the fonts in the FACS plots are too small to be readable.

Reviewer #3: (No Response)

PLOS authors have the option to publish the peer review history of their article (what does this mean?). If published, this will include your full peer review and any attached files.

Reviewer #1: No

Reviewer #2: No

Reviewer #3: No
---

## [Editor Report · Decision Letter 1]

24 Jun 2021

Dear Dr CARDINAUD,

We are pleased to inform you that your manuscript 'Targeting human langerin promotes HIV-1 specific humoral immune responses' has been provisionally accepted for publication in PLOS Pathogens.

Best regards,

Alexandra Trkola

Section Editor

PLOS Pathogens

Thomas Hope

Section Editor

PLOS Pathogens

Kasturi Haldar

Editor-in-Chief

PLOS Pathogens

orcid.org/0000-0001-5065-158X

Michael Malim

Editor-in-Chief

PLOS Pathogens

orcid.org/0000-0002-7699-2064
---

## [Editor Report · Acceptance letter]

26 Jul 2021

Dear Dr CARDINAUD,

We are delighted to inform you that your manuscript, "Targeting human langerin promotes HIV-1 specific humoral immune responses," has been formally accepted for publication in PLOS Pathogens.

Best regards,

Kasturi Haldar

Editor-in-Chief

PLOS Pathogens

orcid.org/0000-0001-5065-158X

Michael Malim

Editor-in-Chief

PLOS Pathogens

orcid.org/0000-0002-7699-2064